# CNN Compression and Search Using Set Transformations with Width Modifiers on Network Architectures

## Abstract

We propose a new approach, based on discrete filter pruning, to adapt off-the-shelf models into an embedded environment. Importantly, we circumvent the usually prohibitive costs of model compression. Our method, Structured Coarse Block Pruning (SCBP), prunes whole CNN kernels using width modifiers applied to a novel transformation of convlayers into superblocks. SCBP uses set representations to construct a rudimentary search to provide candidate networks. To test our approach, the original ResNet architectures serve as the baseline and also provide the 'seeds' for our candidate search. The search produces a configurable number of compressed (derived) models. These derived models are often 20% faster and 50% smaller than their unmodified counterparts. At the expense of accuracy, the size can become even smaller and the inference latency lowered even further. The unique SCBP transformations yield many new model variants, each with their own trade-offs, and does not require GPU clusters or expert humans for training or design.

## 1 Introduction

Modern Computer Vision (CV) is dominated by the convolution operation introduced by Fukushima & Miyake (1982) and later advanced into a Convolutional Neural Network (CNN or convnet) by LeCun et al. (1989). Until recently, these convnets were limited to rudimentary CV tasks such as classifying handwritten digits LeCun et al. (1998). Present-day convnets have far surpassed other CV approaches by improving their framework to include faster activations Nair & Hinton (2010), stacked convolutional layers (convlayers) Krizhevsky et al. (2012), and better optimizers Kingma & Ba (2014). These multi-layer deep convnets require big data in the form of datasets such as ImageNet Deng et al. (2009) to enable deep learning LeCun et al. (2015) of the feature space.

However effective, convnets are held back by their high resource consumption. Utilizing an effective convnet on the edge presents new challenges in latency, energy, and memory costs Chen & Ran (2019). Additionally, many tasks, such as autonomous robotics, require realtime processing and cannot be offloaded to the cloud. As such. resource constrained platforms, such as embedded systems, lack the compute and memory to use convnets in their default constructions.

Analysis into convnets reveals that they are overparameterized Denil et al. (2013) and that reducing this overparameterization can be a key mechanism in compressing convnets Hanson & Pratt (1988); LeCun et al. (1990); Han et al. (2015a). The many weights that form a network are not necessarily of the same entropy and can therefore be seen as scaffolding to be removed during a compression step Hassibi & Stork (1993); Han et al. (2015b); Tessier et al. (2021).

In this work, our objective is to reduce the size of any given convnet using an automated approach requiring little human engineering and compute resources. To that end, we design Structured Coarse Block Pruning (SCBP), a compressing mechanism that requires no iterative retraining or fine-tuning. SCBP uses a low-cost search method, seeded with an off-the-shelf network, to generate compressed models derivatives with unique accuracy, size, and latency trade-offs.

The reminder of this paper is organized as follows. Section 2 focuses on closely related works. Section 3 details the methodology and implementation of SCBP. Section 4 discusses experimental findings, and finally we conclude with key takeaways and future directions in Section 5.

## 2 RELATED WORKS

Early work on removing parameters from Artificial Neural Networks (ANNs) was focused in gaining insights on the purpose of those parameters Hanson & Pratt (1988). Prior to AlexNet Krizhevsky et al. (2012), exploiting ANN overparameterization was used as a regularization mechanism LeCun et al. (1990); Hassibi & Stork (1993). Recently, ANN overparameterization is exploited to reduce the size of models Han et al. (2015b); Zhou et al. (2017); Tessier et al. (2021). Removing parameters compresses the memory footprint of CV models, which can then allow their deployment on embedded systems. Compression additionally facilitates reduced energy costs while also reducing latency by greatly reducing memory traffic Han et al. (2015a); Zhou et al. (2017).

Model accuracy is sustained or reduced, depending on the method of compression. Preserving a compressed CV model's baseline accuracy is challenging and requires large compute Han et al. (2015b); Zhou et al. (2017). A common mechanism for maintaining a trained model's accuracy is to iteratively reduce its size in prune-retrain cycles. Another mechanism is leveraging a Network Automated Search (NAS), often using reinforcement learning, to build networks from scratch that are both small and accurate Zoph et al. (2018); Cai et al. (2020). However, both prune-retrain and NAS are exorbitant in compute usage, typically on the order of $10^3$ and $10^5$ GPU hours respectively.

When computing resources are limited, faster mechanisms for compression are needed. A range of techniques are available, such as tensor factorization Kim et al. (2015); Phan et al. (2020); Swaminathan et al. (2020) and Fast Fourier Transforms (FFT) on CV models' weight tensors. Hashing can also be used to group similar weights into buckets Chen et al. (2015); Hu et al. (2018). These techniques, while faster to train, do not maintain the original network's accuracy and often produce larger models relative to prune-retrain and NAS approaches.

Quantization is also frequently used Gong et al. (2014); Wu et al. (2016) to reduce the bit-width of parameters from 64-bit floats down-to 5-bit ints or less Wu et al. (2016); Zhou et al. (2017). In special cases, only 2-bit parameters are sufficient Rastegari et al. (2016); Courbariaux et al. (2016; 2015). Other techniques include those based on weight-decay and entropy Luo & Wu (2017); Min et al. (2018); Tessier et al. (2021).

In our proposed mechanism, we bridge a gap between manual and NAS approaches by using a low-cost search to order network width attributes of any given CV model, which is partitioned by a novel algorithm into multiple segments with each being assigned its own width modifier. A close work is from Howard et al. (2017) in the form of MobileNets which are a family of networks using different uniform width modifiers on a manually engineered baseline model. Similarly, EfficientNets Tan & Le (2019) expands the idea of modifiers to include depth and input resolution modifiers. Our approach benefits from generalized compression that can be applied to any model because we do not require a new baseline that needs to be engineered and thus can keep cost within $10^1$ GPU hours.

## 3 METHOD

The compression approach detailed below is realized by novel combination of convlayer binning, width modification, and a unique search-train step based on set transforms of the aforementioned combination. Unlike most network architecture search methods that impose prohibitively long search and train times, our work circumvents the cost problem by providing a halfway point between NAS and human engineered architectures. In doing so, we present a rudimentary proof of concept which, in our evaluations, can produce an efficient search and thus generate derivative models when configured by simple human defined search domain set constraints. The SCBP version we use stands on four foundations: (1) a seed architecture from which derivative architectures will be produced; (2) a network segmentation mechanism for the seed architecture for binning and assist in derivation; (3) a set of compression ratios (c-ratios) for each segment of the seed network; and (4) a one shot search for network instantiation based on (1)-(3).

## 3.1 SEED ARCHITECTURES

The instantiation of compressed, derivative architectures is sourced from a seed architecture. In this work, we use the ResNet family as seed architectures. SCBP initialized with a seed network helps cut down on search times by leveraging already known and working architectures to generate new derived variants. Currently, these variants do require training to convergence to determine their accuracy but their latency, memory footprint, and power stats are immediately known. As an aside, the residual connection paradigm in ResNets is widely used today as the foundation of a variety of architectures. As such, using the ResNet testbed here allows for potential extension to later developed networks.

To further accelerate search, the seed architecture needs to be segmented into three portions, each of which undergoes its own unique compression. ResNet on CIFAR data consists of three *superblocks* where we define a superblock to be all ResNet blocks of the same filter dimensions. Thus, we use three segments because our seed architecture accommodates it with little engineering. The input layer and its subsequent down-sampling layers plus the output layers are left untouched.

Once the seed network is divided into well-defined segments, each segment is selected for width modification by applying c-ratios. Different segment and c-ratio pairings show changed weight distributions and residual functions and hence result in new derived networks. Interestingly, the features learned per superblock, to a certain extent, can change to maintain accuracy when adjacent superblocks are compressed. i.e. Over-parameterized segments may absorb entropy that may be lost from adjacent c-ratios.

To detail, in each derived network, an ordered tuple of c-ratios is required, where each element in the tuple corresponds to a segment and hence encodes the compression factor for that segment. Thus, a set $\mathbf{R}$ of c-ratios can be constructed using the Cartesian product of $\mathbf{R}$ and the set $\mathbf{S}$ of partitioned segments. Both $\mathbf{S}$ and $\mathbf{R}$ must be small countable sets to prevent combinatorial explosion. In section 4, we find excellent results with $|\mathbf{S}| = 3$ and $|\mathbf{R}| = 4$. If a seed architecture can potentially benefit from different $\mathbf{S}$ and $\mathbf{R}$, these hyperparameters can be easily changed.

The emergent property from the architecture derivation process above is an effective and quick representation component for enumerating architectures that circumvents intractable search costs. The simplicity of the method allows its application to the growing library of modern convnets. It is therefore possible for many of these off-the-shelf convnets to be automatically modified via compression to meet embedded system resource constraints. This provides a low-cost approach to leverage past and present architecture engineering effort in embedded use cases.

## 3.2 SELECTING AND APPLYING C-RATIOS TO NETWORK SEGMENTS

A segment is a binning of sequentially stacked convolutional layers of the same dimensions. These segments are craved up from the seed network. The number of segments in a seed network is dependent on two factors: the seed's architecture and the segmenting procedure. For ResNet like models, segmenting is straight-forward: use each superblock as its own segment. In different architectures, such as VGG-19, segmenting can done based on like dimension convolutional layers. The procedure should be adapted based on the seed architecture, the cardinality of $\mathbf{R}$, and available compute. In algorithm 1, we provide a generalized segmenting procedure for arbitrary convnets.

Convlayers can be represented in the ordered set $\mathbf{C}$ and it is from here that $\mathbf{S}$ is constructed. Both $\mathbf{S}$ and $\mathbf{C}$ are in the same set family. A one-to-one correspondence between $\mathbf{S}$ and $\mathbf{C}$ is to be avoided, which would mean each convlayer is its own segment. A small segment count is crucial to an efficient search because it limits the number of generated derived networks, as described in equation 4. We can understand each segment $s \in \mathbf{S}$ as a coarse representation of multiple $c \in \mathbf{C}$. In practice, algorithm 1 implements the following superblock segment definition:

$$\mathbf{S} = \{s : (c_1, \dots, c_k) \in \mathbf{C}, \ |s| \in \mathbb{N}, \ 1 \leqslant k, |s| \leqslant |\mathbf{C}|\} \tag{1}$$

where each $s$ is a tuple of multiple $c$. The cardinality of $\mathbf{S}$ is not required to be identical to $\mathbf{C}$.

After network segmentation into superblocks, we need to pair these segments with a c-ratio. The c-ratio, at its core, is a multiplicative factor applied to the filter dimension of convlayers. It is a number strictly $\leqslant 1.00$. All baseline convolutional layers are at a default c-ratio of 1.00, and this is a valid configuration for SCBP. The c-ratio $\mathbf{r}$ is constructed from the set $\mathbf{R}$ as follows:

$$\mathbf{R} = \{r : r \in \mathbb{R} \quad and \quad 0.0 < r \leqslant 1.00\} \tag{2}$$

where **r** values <1.0 compress convlayers by discrete filter pruning (truncation), $r$=1.0 preserves the original convlayer width attributes, and c-ratios >1.00 are undefined in terms of compression. The operation of applying a c-ratio to a segment is essentially a transform T such that:

$$T(m \times n \times f \times k) = m \times n \times \lceil (f \times \mathbf{r}) \rceil \times k \tag{3}$$

Here m $\times$ n represent the filter's spacial dimensions, and $f$ is the filter count. $f$ is modified by **r**, and k denotes the segment depth (simply, its the stacked convlayer count).

Selecting the correct c-ratios can both be done using a search, such as a grid-search, or manually. Our experiments in section 4 use four manually selected c-ratios. In our experience, SCBP shows resiliency and works well without fine-tuned c-ratios. With that said, a low-cost fine-tuning step for c-ratios is a possible future direction. In this work's seed networks, the four c-ratios were hand-picked as equidistant points from one another to provide a coarse, uniform coverage of the c-ratio search space. They are the set **R** = {0.25, 0.50, 0.75, 1.00}.

The segment and c-ratio sets are used in conjunction to produce compressed *derived* networks. Derived networks can be thought of as subnets of their overparameterized seed networks. Each subnet behaves uniquely, due in part to their differing weight distribution. The construction of the collection of derived networks requires both sets **R** and **S**. Each tuple $r \in \mathbf{R}$ is crossed with a seed segment $s \in \mathbf{S}$ using the Cartesian product between the sets:

$$\prod_{i=1}^{|S|} \mathbf{R}_i \bigtimes \mathbf{S} = \{(s_1 r_1, \dots, s_n r_n) : \forall_s \exists_r where \ r \in \mathbf{R}, \ s \in \mathbf{S}, \ and \ n = |S|\} \tag{4}$$

Each segment has its own c-ratio where the c-ratio set is transformed into a multiset with multiplicity equal to the segment set cardinality. The cross product between the c-ratio multiset and segment set yield the space of derived networks. As such, each derived network is configured with unique memory footprints and weight tensors; these configurations can then allow similarly accurate networks to be profiled and culled for the best hardware fit.

There are no prune-retrain cycles and hyperparameters are simply adopted from the seed network. The derived network tensors are not sparse and this non-sparsity is of great benefit; it allows hardware acceleration of multiply-add on virtually all platforms without the need of ASICs.

### 3.3 APPLYING SCBP TO RESNETS AND OTHER ARCHITECTURES

CV models typically utilize stacked convlayers, where many successive layers are of identical dimensions. Residual connections are also common in these modern convnets. While innumerable improvements have been made in activations, bottleneck layers, etc, the underlying data structures of most contemporary models stands on the foundations of convlayers and residual connections.

Algorithm 1 forms the basis of SCBP as it segments the seed network into superblocks by binning like-dimension convlayers. The bins are sorted according to parameter count where the largest $k$ bins are selected as segments and k is configurable hyperparameter (the segment set cardinality).

| **Algorithm 1:** segmenting procedure | **Algorithm 2:** derivation procedure |
|---|---|
| seed ← select_convnet(model_constraints);
layers ← extract_dims(seed);
buckets ← bin_like_dims(layers);
s_buckets ← sort_descending(buckets);
segments ← s_buckets[0...k];
return segments; | segments ← segmenting_procedure();
cratios ← multiset_transform(cratio_set, multiplicity
  ← segments.length);
derived_segments ← cartesian_product(segments,
  cratios);
return derived_segments; |

To create derived segments from a binned seed network, we must first identify a set of c-ratios. In this work, we use a default set, given in algorithm 2, for two reasons. One, no additional search is incurred,

and second, the c-ratios and their magnitude evenly covers the subspace of widths not larger than the baseline width. We exploit the block structure of our seed architecture by coalescing blocks into superblocks and then selectively compressing these superblocks such that non-uniform compression can be performed to derive both networks and insights into the original seed. Additionally, limiting c-ratios to superblocks (instead of convlayers) significantly cuts down on the enumeration cost for the architecture search. The final derived segments are a product of both the binned convlayers and c-ratios. This product is the building unit of each derived network.

The network generation for derived networks starts at the seed network and ends with a pool of derived networks, as given in 3. The procedure acquires a seed, creates segments from the seed and then pairs the segments with c-ratios to construct derived segments. The derived segments are then pieced back together to form the new derived networks.

---

**Algorithm 3:** network generation

seed ← select_convnet(model_constraints);
segments ← segmenting_procedure();
derived_segments ←
  derivation_procedure(segments);
derived_networks ← list();
**for** *segment_tuple in derived_segments* **do**
  template ← seed.deepcopy();
  network ← replace_layers(template,
    segment_tuple);
  derived_networks.append(network);
**end**
return derived_networks;

---

**Algorithm 4:** train, test, rank

derived_networks ← network_generation();
models ← list();
performance ← list();

**for** *convnet in derived_networks* **do**
  model ← train(convnet);
  metrics ← test(model);
  models.append(model);
  performance.append(metrics);
**end**

candidate_models ← rank(models,
  performance);

---

The train-test regiment remains the same as the seed, but instead of one network, many more networks are trained in parallel. Each derived model has its metrics logged for later ranking so that the best one can be determined based on user-specific constraints. In this work, our c-ratio and segmenting procedures construct 64 unique models per seed. We random initialize each model, train-validate it, and finally collect metrics on the testset.

SCBP is generalizable to most deep convnet architectures that use sequentially stacked convlayers. However, when the segmenting procedure is unable to bin layers, as is the case when all convlayers are of different dimensions, SCBP can become limited, especially when segments cover a minor subset of all the parameters. In practice, modern convnets typically use many stacked convolutional layers and thus are receptive to SCBP.

In sum, SCBP can generate compressed models without the need to modify training regiments, continuous fine-tuning, prune-retrain cycles, or even modified hyperparameters. The compression of superblocks to different c-ratios can provide valuable insights into their relative importance to the rest of the network. These insights can guide efforts to build better architectures and may also help find hardware quirks that adapt better to certain derived architectures.

## 4  EVALUATIONS AND DISCUSSIONS

### 4.1  SETUP, METRICS, AND EXPERIMENTS

The CIFAR dataset is used to determine the effectiveness of SCBP on the compressed models. CIFAR provides a complex data distribution with coarse labels that are appropriate for approximating tasks on the edge. The dataset consists of 50K train images and 10K test images. In total, these 60K images, of dimensions 32x32x3, cover a variety of categories such as dogs and trucks. The dataset has two label sets over the same data, one with 10 coarse labels and another with 100 finer labels. We bench with on both using the top-1 metric.

Inference statistics are measured on the *JETSON Xavier NX* and a desktop *WorkStation*. The platforms serve to approximate real-world hardware constraints so that the effects of compression can be evaluated in terms of latency and power. In table 3, the *WorkStation* is the least constraint while low memory bottlenecks while the *Xavier NX* provides a typical embedded environment for benching.

| sblock1 | sblock2 | sblock3 |
|---------|---------|---------|
| 5.196%  | 18.935% | **75.455**% |

Table 1: **Parameter distribution.** The density of weights in convnets is usually spread nonuniformly. For example, our ResNet testbed has almost all parameters are concentrated into the last two superblocks. Compression on high density superblocks yields faster and smaller models while identical c-ratios on lower density superblocks are correlated with worse accuracy.

| hyperparameters | | | |
|-------|-------|----------|---------------|
| decay | epoch | momentum | learning rate |
| 1e-4  | 1e2   | 9e-1     | 1e-1          |

Table 2: **Hyperparameters.** Every derived network repurposes the seed network's hyperparameters. Here, the training regiment is adapted from He et al. (2016) with no modifications to demonstrate the applicability of SCBP. Training and testing is done using PyTorch 1.9v with two RTX 2080Ti GPUs.

| | Platform | |
|---------------|----------|-------------|
| | XavierNX | WorkStation |
| Component     |          |             |
| L1 Cache      | 1MB      | 512KB       |
| L2 Cache      | 6MB      | 2MB         |
| L3 Cache      | 4MB      | 16MB        |
| Total Cache   | 11MB     | 18.5MiB     |
| CPU Memory    | 8GB      | 64GB        |
| Video Memory  | 8GB      | 11GB        |
| Total Memory  | 8GB      | 75GB        |
| CPU Cores     | 6        | 8           |
| CUDA Cores    | 384      | 4352        |

Table 3: **Hardware.** The platforms above each have significantly differing computation capacities which help demonstrate the effects of compression rates on different resource constraints. Specifically, both the embedded JETSON and Raspberry Pi use shared memory architectures for their gpu and cpu cores. The WorkStation uses dedicated graphics memory. All the platforms have different cache associativities, clocks speeds, and memory latencies.

Performance is composed of several metrics. In our experiments, we settle on four measures of performances: accuracy, latency, size, and power. These metrics help determine a model's holistic fit into an embedded environment. Top-1 accuracy is measured using the whole held-out testset. The model size is tied to superblock c-ratios and table 1 provides a reference. Latency and energy-interval are dynamic metrics that need averaging over many readings to reduce their margin of error. Both the model latency and power draw are processed as arithmetic means from 10k inference frames.

The seed architectures are unadulterated models pulled from He et al. (2016). We use ResNet20, ResNet44, and ResNet110 to provide 'source' candidates for the SCBP based search. Additionally, using ResNets helps to indirectly approximate their innumerable successor networks, many of which are minor architectural changes and most of which are prevalent in production today. For CIFAR100 labels (new models), the final denselayer of the ResNets are modified to simply increase node count from 10 to 100. All models use zero-padded summation for their residual connections.

## 4.2 EFFECT OF SCBP ON ACCURACY AND MODEL SIZE

This section provides an overview of the empirical effects of SCBP and further analyzes its effects on model architecture and performance. In particular, SCBP reveals interesting architecture patterns that we discuss and highlight.

Parameter updates during training change based on the label granularity. That is, the superblock-to-accuracy relation begins to shift with just a change in the number of labels. When controlling for parameter count, coarse labels on shallow networks usually outperform deeper networks. Conversely, deeper networks outperform shallow networks on finer labels, even when parameter counts are similar. Label granularity additionally influences the distribution of redundant parameters, where coarse labels primarily increase redundancy in the final superblock and fine labels distribute low entropy parameters across all superblocks see table 5. Interestingly, c-ratios have similar accuracy effects, regardless of superblock filter density. For example, a c-ratio of 0.25 on the first superblock usually harms accuracy much more than on the same c-ratio on the significantly denser last superblock. These patterns are evident from experiments, especially with derived nets seeded with ResNet44, see table 4. The tables also indicate the learning ability of different portions within the seed networks by evaluating their response to SCBP.

In terms of accuracy resilience to truncations, both the learning task and cumulative layers up to a given superblock need to be considered. The compression impact on accuracy is better correlated with layer count and data granularity than the actual parameter count. For example, the finer-grained labels

Figure 1: These plots demonstrate the effect of c-ratios {0.25,0.50,0.75,1.00} on superblocks one and two when superblock three is fixed to {0.25,0.50}. The z-axis is the delta deviation in accuracy of the derived networks from their respective seed baselines. Like most convnets, parameters are concentrated at the tail end and hence aggressive compression, such as with a c-ratio <=0.5, is of interest. Note that more compression has the effect of fewer derived models meeting the accuracy threshold.

| CIFAR10 | sblock1 | sblock2 | sblock3 |
|---|---|---|---|
| cr ← 0.25 | 15 | 9 | 13 |
| cr ← 0.50 | 26 | 23 | 23 |
| cr ← 0.75 | 30 | 32 | 31 |
| cr ← 1.00 | 32 | 39 | 36 |

| CIFAR100 | sblock1 | sblock2 | sblock3 |
|---|---|---|---|
| cr ← 0.25 | 11 | 4 | 0 |
| cr ← 0.50 | 13 | 9 | 9 |
| cr ← 0.75 | 13 | 15 | 14 |
| cr ← 1.00 | 12 | 21 | 26 |

Table 4: **CIFAR10 trends.** The table shows the number of derived models (out of 192) that meet an accuracy delta of $\pm 2\%$ from the seed's baseline accuracy. Each table cell encodes the given superblock locked to the given c-ratio and all other superblocks are free variables. In conjunction with table 1, we know superblock one only has 5% of parameters but it can disproportionally effect candidate net count with c-ratio=0.25. Similar insights guide us to instead use derived nets with superblock three at c-ratio=0.25 and accumulate a similar number of candidates with a size reduction of 19% instead of just 5%. Of the 192 derived networks, 103 meet the accuracy constraint.

Table 5: **CIFAR100 trends.** Unlike table 4, the more 10X more labels in the dataset makes learning difficult as seen in only 49 derived networks meeting the $\pm 2\%$ accuracy constraint, a 52.4% reduction from the CIFAR10 result. Interestingly, the pattern of learning per superblock seems to change as reflected in the table columns. Unlike previously, superblock one stays stable regardless of c-ratio, and now the other blocks suffer a 50% drop in candidate model counts for each successive c-ratio step. In short, it may be possible to significantly increase compression by reducing label granularity in preprocessing. However, the generality and feasibility of this approach needs to be developed.

in CIFAR100 make compression more challenging because each filter encodes more information and therefore incurs a larger cost when it is pruned. This can be inferred from table 4 and figure 1 where the simple CIFAR10 labels allow SCBP to produce 110% more derived networks than with CIFAR100's finer labels.

Within the derived networks, we note several learning patterns. One such pattern is that the final superblock, the largest in our experiments, counter-intuitively contributes the least to final model accuracy. In fact, even removing 50% to 75% of its filters incurs marginal accuracy effects. The large truncation allowance on the final superblock is beneficial because it means that many derived models can be compressed 40-50%.

Another pattern SCBP exposes is a weak co-dependence between the first two superblocks where many models lose >2% accuracy if 75% of their parameters are truncated in either superblock.

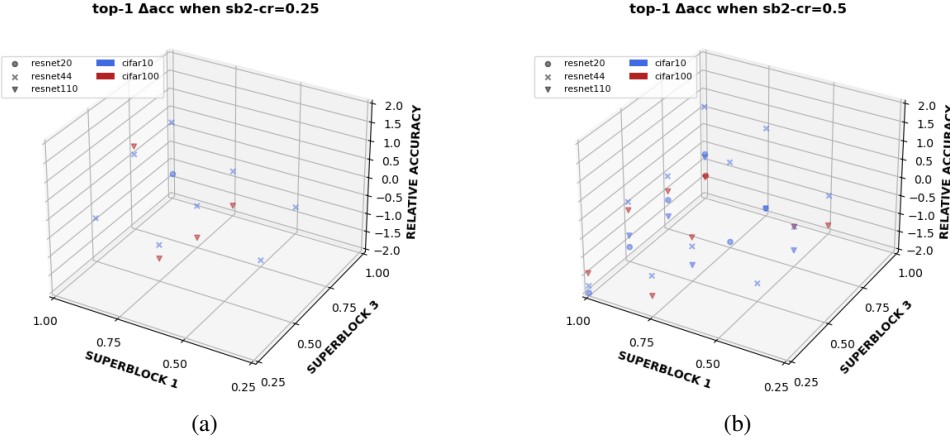

(a)                                                    (b)

Figure 2: Here, instead of superblock three, we fix superblock two's c-ratio into {0.25,0.50} and leave the other superblocks free within {0.25,0.50,0.75,1.00}. Unlike the previous 1, there are many more derived nets that meet the accuracy threshold. Superblock two has significantly less parameters, about 19% of the total. See table 1 for parameter distributions. The overall network compression is modest here, even with aggressive c-ratios. The noteworthy trend is significantly more derived networks meeting the accuracy threshold with many at or above the seed network's baseline.

Extensive data tables are provided in Appendix 5. The co-dependence may be due to the low parameter redundancy present in the first two superblocks. Indeed, these possibly higher entropy shallow superblocks may indicate that overparameterization is mainly a problem of deeper layers. The relative compression-accuracy trade-off between superblocks means its better to focus on compressing latter superblocks. Additionally, it should be noted from table 6 and figures **??** that the middle superblock has a large facility to absorb critical weights from its neighbors when they are highly compressed. More generally, high compression tends to cause adjacent superblocks to compensate for missing filters. After a certain amount, typically 30% size reduction, lowering model size correlates positively with lowering accuracy, albeit with some derived net exceptions that are more accurate after compression (see ResNet44-1.0-0.5-1.0, etc).

Our takeaway is that SCBP as applied to convnets can effectively compress models with very low search costs. The c-ratios on superblocks help illuminate the relative contribution of parameters to final performance and therefore provide insights into the seed and can possibly assist in designing more compressible seed networks. And while SCBP size reductions are moderate when compared to unstructured pruning, SCBP does not result in weight matrix sparsity and thus benefits from BLAS-based hardware acceleration. Lastly, SCBP saves time by not requiring hyperparameter tuning or modification to the seed's training regiment. These attributes coupled with no prune-retrain cycles, elevate SCBP far above many comparable techniques in terms of GPU costs. Typically, these costs are in the range of $10^3$ hours for pruning and up-to $10^5$ hours for NAS. In comparison, SCBP consumes less than 72 GPU hours from start to finish.

### 4.3 Effect of SCBP on Latency and Energy

Inference time is end-to-end, meaning it encompasses image preprocessing, data moves, model execution, and postprocessing. It is this time that is averaged for the latency and its during this duration that power draw is also measured. For power draw, the *Xavier NX* polls its INA3321 micro-controller while the *WorkStation* samples from *turbostat* and *nvidia-smi*.

The main determining factor for latency and power metrics is model size. No matter the architecture or hardware, frequent cache misses and, worse, memory swaps absolutely crush real-time performance and balloon latency. The speed and capacity of the memory hierarchy is the determining factor of latency and hence the energy-interval. Because the main contributor to latency and energy costs is memory access, designing smaller convnets is an effective approach for resource constrained platforms which heavily benefit from compression due to their limited memory hierarchy.

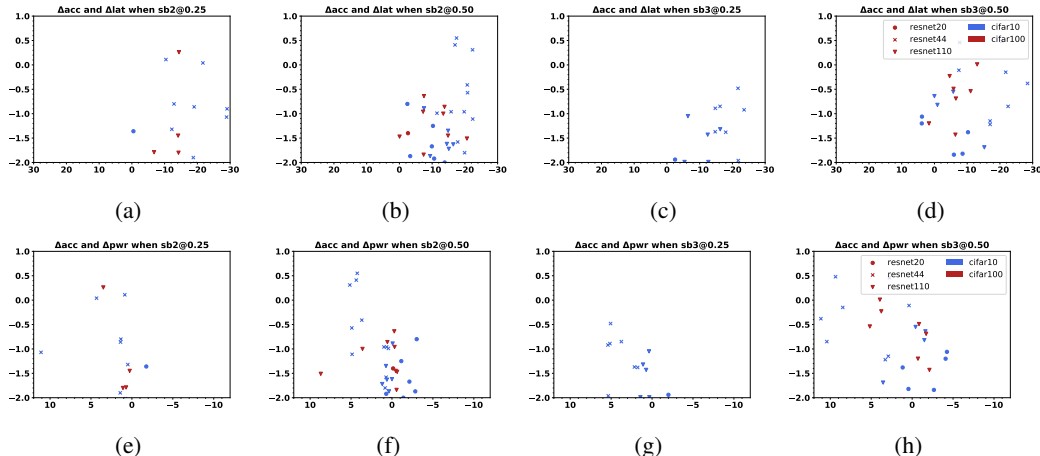

Figure 3: **XavierNX General Trends in Latency & Power** The x-axis is percentage change, centered at 0, where less is better (to the right). The y-axis shows the delta in accuracy from each seed ResNet's respective baseline (positive and higher is better). The stellar pattern to note is that of latency in subfigs (a)-(d) where we see greatly reduced latencies, usually by over 20% while accuracy is largely maintained. Power draw, as seen in (e)-(h) remains steady with little delta deviation from the baseline.

In figure 1, we see that derived nets can maintain their accuracy while compressed. This means there are smaller models, with possibly fewer cache misses, which could be faster. Then we see in figures 3, (a)-(d), that indeed latency is greatly reduced, by around 20%, and 30% in some cases. Meanwhile, the actual power draw seen in figures 3 (e)-(h) does not show significant changes meaning that overall energy-interval are reduced for our derived networks.

Lastly, we find that once a memory bottleneck is successfully eliminated, further compression comes with diminishing returns as can be observed in the WorkStation experiments in the Appendix, figure 4. The data indicates that excessive parameter pruning, past memory bottlenecks is an unneeded computation waste. The pruning, as deployed in many unstructured compression techniques does not lead to faster, less energy hungry models; often they only result in smaller models coming at the cost of very expensive and long train cycles.

## 5 CONCLUSION AND FUTURE WORK

We demonstrate a novel mechanism to segment convlayers into superblocks and set them with different compression ratios using a set represented network search. The SCBP framework constructs a pool of models with different attributes which can help with different hardware fits. These models are much smaller and faster than their unmodified counterparts. The training cost of these models is low and feasible with regular workstations. As such, the embedded native models can be designed without prohibitive costs, allowing rapid iteration to find the best model-hardware pairing. For future work, we are extending SCBP into an iterative mode that operates on pre-trained networks.

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

WORKSTATION BENCHMARKS

Unlike figure 3, neither power nor latency show significant deviation from the seed network's baseline, hinting at the diminishing return of pruning once bottlenecks are no longer present.

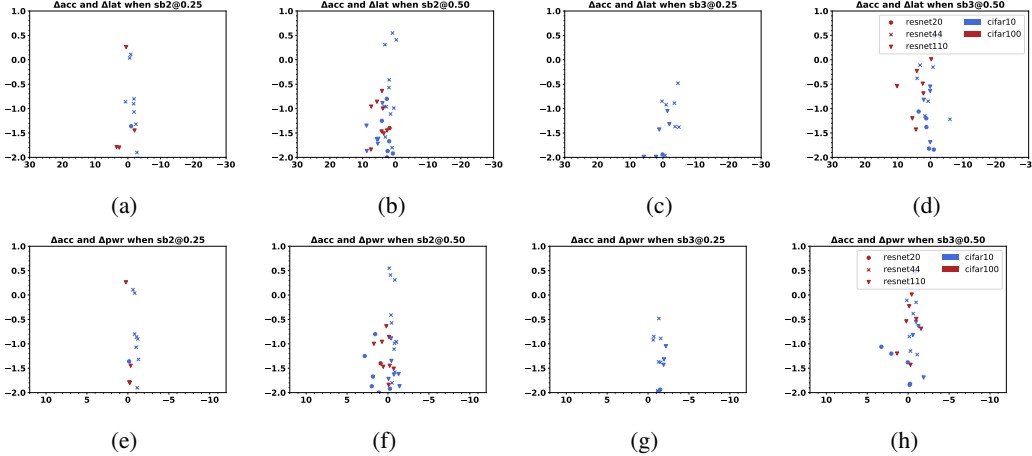

Figure 4: **WorkStation General Trends in Latency & Power** Similar to figure 3, the x-axis and y-axis are measure percentage changes.

ACCURACY AND MEMORY TABLES

The complete testing results for the derived networks are given in the tables below. Each table data cell represents a single derived model's metrics with the following representation: $\delta\mathrm{acc}_{sz}$. The accuracy (acc) of each model is the delta from its seed network. The size (sz) is given as a *percentage* of the seed's size; i.e. 0.6 means 60% of the seed's size or a 40% size reduction for that particular derived convnet.

The row and column heading cells denote the superblock c-ratios. Here the c-ratios are given in the form $\{0.25, 0.50, 0.75, 1.00\}^{sb\#}$ where $0.75^{sb2}$ evaluates to superblock two compressed with a c-ratio of 0.75. Each table row iterates over the c-ratio set, from 0.25 to 1.00, modulus 4. The columns are layered into two levels; at the top is superblock three, e.g. $0.50^{sb3}$, and this only changes every four columns. Below the superblock three heading is superblock two configuration, and it iterates similar to the rows, i.e. each column is difference c-ratio on superblock two.

**ResNet20**

| | $0.25^{b3}$ | | | | $0.50^{b3}$ | | | | $0.75^{b3}$ | | | | $1.00^{b3}$ | | | |
|---|---|---|---|---|---|---|---|---|---|---|---|---|---|---|---|---|
| | $0.25^{b2}$ | $0.50^{b2}$ | $0.75^{b2}$ | $1.00^{b2}$ | $0.25^{b2}$ | $0.50^{b2}$ | $0.75^{b2}$ | $1.00^{b2}$ | $0.25^{b2}$ | $0.50^{b2}$ | $0.75^{b2}$ | $1.00^{b2}$ | $0.25^{b2}$ | $0.50^{b2}$ | $0.75^{b2}$ | $1.00^{b2}$ |
| $0.25^{b1}$ | $-9.90_{.09}$ | $-7.00_{.13}$ | $-4.97_{.32}$ | $-3.97_{.40}$ | $-5.55_{.22}$ | $-4.01_{.26}$ | $-3.12_{.54}$ | $-2.12_{.63}$ | $-4.51_{.43}$ | $-3.14_{.48}$ | $-1.99_{.84}$ | $-1.52_{.93}$ | $-3.23_{.73}$ | $-2.00_{.78}$ | $-1.64_{.26}$ | $-1.16_{.10}$ |
| $0.50^{b1}$ | $-8.22_{.10}$ | $-6.03_{.14}$ | $-4.30_{.2}$ | $-3.43_{.28}$ | $-4.74_{.23}$ | $-3.73_{.27}$ | $-2.19_{.34}$ | $-1.84_{.42}$ | $-2.84_{.44}$ | $-2.09_{.49}$ | $-1.36_{.56}$ | $-0.78_{.64}$ | $-2.70_{.74}$ | $-1.67_{.79}$ | $-0.98_{.86}$ | $-0.33_{.95}$ |
| $0.75^{b1}$ | $-6.94_{.12}$ | $-5.25_{.16}$ | $-3.89_{.22}$ | $-2.57_{.30}$ | $-3.59_{.25}$ | $-2.52_{.29}$ | $-1.82_{.35}$ | $-1.38_{.44}$ | $-2.47_{.46}$ | $-1.92_{.51}$ | $-1.18_{.57}$ | $-0.89_{.66}$ | $-2.06_{.75}$ | $-1.25_{.81}$ | $-0.66_{.88}$ | $-0.23_{.97}$ |
| $1.00^{b1}$ | $-5.99_{.14}$ | $-4.37_{.18}$ | $-3.02_{.24}$ | $-1.94_{.32}$ | $-3.39_{.27}$ | $-2.17_{.32}$ | $-1.20_{.38}$ | $-1.06_{.47}$ | $-2.19_{.48}$ | $-1.87_{.53}$ | $-1.18_{.60}$ | $-0.08_{.69}$ | $-1.36_{.78}$ | $-0.80_{.83}$ | $-0.65_{.90}$ | $0.00_{1.0}$ |

**ResNet44**

| | $0.25^{b3}$ | | | | $0.50^{b3}$ | | | | $0.75^{b3}$ | | | | $1.00^{b3}$ | | | |
|---|---|---|---|---|---|---|---|---|---|---|---|---|---|---|---|---|
| | $0.25^{b2}$ | $0.50^{b2}$ | $0.75^{b2}$ | $1.00^{b2}$ | $0.25^{b2}$ | $0.50^{b2}$ | $0.75^{b2}$ | $1.00^{b2}$ | $0.25^{b2}$ | $0.50^{b2}$ | $0.75^{b2}$ | $1.00^{b2}$ | $0.25^{b2}$ | $0.50^{b2}$ | $0.75^{b2}$ | $1.00^{b2}$ |
| $0.25^{b1}$ | $-6.77_{.09}$ | $-4.20_{.12}$ | $-3.49_{.18}$ | $-1.38_{.26}$ | $-3.62_{.22}$ | $-2.44_{.26}$ | $-1.15_{.32}$ | $-1.22_{.40}$ | $-3.48_{.45}$ | $-0.96_{.48}$ | $-0.73_{.55}$ | $-1.69_{.63}$ | $-2.49_{.76}$ | $-1.80_{.80}$ | $-0.60_{.86}$ | $-1.10_{.94}$ |
| $0.50^{b1}$ | $-5.20_{.10}$ | $-2.55_{.13}$ | $-1.96_{.19}$ | $-1.37_{.27}$ | $-2.02_{.23}$ | $-1.58_{.27}$ | $-0.38_{.33}$ | $-0.15_{.41}$ | $-1.07_{.46}$ | $-1.11_{.49}$ | $-0.10_{.56}$ | $-0.61_{.64}$ | $-0.86_{.77}$ | $-0.41_{.81}$ | **$+0.30_{.87}$** | **$+0.02_{.96}$** |
| $0.75^{b1}$ | $-3.55_{.11}$ | $-2.13_{.15}$ | $-0.92_{.21}$ | $-0.48_{.29}$ | $-1.90_{.25}$ | $-0.96_{.29}$ | $-0.11_{.35}$ | **$+0.48_{.43}$** | $-0.90_{.47}$ | **$+0.31_{.51}$** | $-0.39_{.57}$ | **$+0.62_{.66}$** | **$+0.04_{.78}$** | $-0.57_{.82}$ | **$+0.45_{.89}$** | $-0.14_{.97}$ |
| $1.00^{b1}$ | $-3.22_{.13}$ | $-2.08_{.17}$ | $-0.85_{.23}$ | $-0.89_{.31}$ | $-1.32_{.27}$ | $-0.99_{.31}$ | $-0.85_{.37}$ | **$+0.46_{.45}$** | $-0.80_{.49}$ | **$+0.41_{.53}$** | **$+0.33_{.60}$** | **$+1.21_{.68}$** | **$+0.11_{.80}$** | **$+0.55_{.85}$** | **$+1.43_{.91}$** | $0.00_{1.0}$ |

**ResNet110**

| | $0.25^{b3}$ | | | | $0.50^{b3}$ | | | | $0.75^{b3}$ | | | | $1.00^{b3}$ | | | |
|---|---|---|---|---|---|---|---|---|---|---|---|---|---|---|---|---|
| | $0.25^{b2}$ | $0.50^{b2}$ | $0.75^{b2}$ | $1.00^{b2}$ | $0.25^{b2}$ | $0.50^{b2}$ | $0.75^{b2}$ | $1.00^{b2}$ | $0.25^{b2}$ | $0.50^{b2}$ | $0.75^{b2}$ | $1.00^{b2}$ | $0.25^{b2}$ | $0.50^{b2}$ | $0.75^{b2}$ | $1.00^{b2}$ |
| $0.25^{b1}$ | $-5.21_{.09}$ | $-3.01_{.12}$ | $-3.48_{.18}$ | $-2.44_{.26}$ | $-3.71_{.22}$ | $-2.85_{.26}$ | $-4.32_{.32}$ | $-1.20_{.40}$ | $-2.26_{.45}$ | $-7.35_{.49}$ | $-2.86_{.55}$ | $-1.95_{.63}$ | $-5.50_{.77}$ | $-3.40_{.81}$ | $-1.81_{.87}$ | $-1.85_{.95}$ |
| $0.50^{b1}$ | $-4.34_{.1}$ | $-2.70_{.13}$ | $-2.36_{.19}$ | $-0.94_{.27}$ | $-2.35_{.23}$ | $-2.59_{.27}$ | $-2.35_{.33}$ | $-2.41_{.41}$ | $-2.74_{.46}$ | $-1.14_{.50}$ | $-1.31_{.56}$ | $-1.50_{.64}$ | $-2.56_{.78}$ | $-0.86_{.82}$ | $-1.44_{.88}$ | $-0.98_{.96}$ |
| $0.75^{b1}$ | $-3.26_{.11}$ | $-2.48_{.15}$ | $-1.50_{.20}$ | $-0.83_{.29}$ | $-2.34_{.25}$ | $-1.13_{.28}$ | $-1.83_{.34}$ | $-0.06_{.42}$ | $-1.72_{.48}$ | $-2.43_{.51}$ | $-1.79_{.57}$ | $-1.01_{.65}$ | $-4.08_{.79}$ | $-1.23_{.83}$ | $-0.52_{.89}$ | $-0.59_{.97}$ |
| $1.00^{b1}$ | $-2.90_{.13}$ | $-2.52_{.17}$ | $-1.50_{.23}$ | $-0.56_{.31}$ | $-1.88_{.27}$ | $-2.61_{.31}$ | $-0.33_{.36}$ | $-0.15_{.45}$ | $-3.99_{.50}$ | $-1.38_{.53}$ | $-5.06_{.59}$ | $-0.81_{.68}$ | $-2.08_{.81}$ | $-0.40_{.85}$ | $0.00_{.91}$ | $0.00_{1.0}$ |

Table 6: **Accuracy & Memory on CIFAR10.** Each subtable lists all derived models for the denote table heading seed convnet.

**ResNet20**

| | $0.25^{b3}$ | | | | $0.50^{b3}$ | | | | $0.75^{b3}$ | | | | $1.00^{b3}$ | | | |
|---|---|---|---|---|---|---|---|---|---|---|---|---|---|---|---|---|
| | $0.25^{b2}$ | $0.50^{b2}$ | $0.75^{b2}$ | $1.00^{b2}$ | $0.25^{b2}$ | $0.50^{b2}$ | $0.75^{b2}$ | $1.00^{b2}$ | $0.25^{b2}$ | $0.50^{b2}$ | $0.75^{b2}$ | $1.00^{b2}$ | $0.25^{b2}$ | $0.50^{b2}$ | $0.75^{b2}$ | $1.00^{b2}$ |
| $0.25^{b1}$ | $-26.58_{.09}$ | $-21.09_{.13}$ | $-17.49_{.19}$ | $-15.44_{.26}$ | $-13.25_{.23}$ | $-9.85_{.27}$ | $-8.18_{.33}$ | $-7.35_{.41}$ | $-7.53_{.44}$ | $-6.06_{.48}$ | $-3.86_{.55}$ | $-2.96_{.63}$ | $-5.79_{.73}$ | $-3.02_{.78}$ | $-1.94_{.85}$ | $-1.77_{.94}$ |
| $0.50^{b1}$ | $-23.02_{.11}$ | $-20.19_{.14}$ | $-16.83_{.2}$ | $-14.07_{.28}$ | $-11.21_{.24}$ | $-9.43_{.28}$ | $-7.33_{.34}$ | $-7.26_{.42}$ | $-6.37_{.45}$ | $-5.81_{.49}$ | $-3.66_{.56}$ | $-3.19_{.64}$ | $-3.94_{.74}$ | $-3.11_{.79}$ | $-1.91_{.86}$ | $-0.89_{.95}$ |
| $0.75^{b1}$ | $-21.61_{.12}$ | $-18.14_{.16}$ | $-15.37_{.22}$ | $-13.27_{.30}$ | $-10.50_{.25}$ | $-8.23_{.3}$ | $-7.06_{.36}$ | $-5.14_{.44}$ | $-6.00_{.47}$ | $-3.73_{.51}$ | $-2.97_{.58}$ | $-2.67_{.66}$ | $-3.27_{.76}$ | $-2.03_{.81}$ | $-1.48_{.88}$ | $-0.13_{.97}$ |
| $1.00^{b1}$ | $-20.14_{.14}$ | $-17.03_{.18}$ | $-15.64_{.24}$ | $-12.93_{.32}$ | $-9.16_{.28}$ | $-7.50_{.32}$ | $-5.57_{.38}$ | $-5.22_{.47}$ | $-4.95_{.49}$ | $-4.24_{.54}$ | $-2.32_{.60}$ | $-2.61_{.69}$ | $-2.74_{.78}$ | $-1.40_{.83}$ | $-0.53_{.90}$ | $0.00_{1.0}$ |

**ResNet44**

| | $0.25^{b3}$ | | | | $0.50^{b3}$ | | | | $0.75^{b3}$ | | | | $1.00^{b3}$ | | | |
|---|---|---|---|---|---|---|---|---|---|---|---|---|---|---|---|---|
| | $0.25^{b2}$ | $0.50^{b2}$ | $0.75^{b2}$ | $1.00^{b2}$ | $0.25^{b2}$ | $0.50^{b2}$ | $0.75^{b2}$ | $1.00^{b2}$ | $0.25^{b2}$ | $0.50^{b2}$ | $0.75^{b2}$ | $1.00^{b2}$ | $0.25^{b2}$ | $0.50^{b2}$ | $0.75^{b2}$ | $1.00^{b2}$ |
| $0.25^{b1}$ | $-20.03_{.09}$ | $-17.50_{.12}$ | $-13.75_{.18}$ | $-12.16_{.26}$ | $-9.69_{.23}$ | $-7.89_{.26}$ | $-5.60_{.32}$ | $-6.38_{.40}$ | $-6.69_{.45}$ | $-4.85_{.49}$ | $-4.31_{.55}$ | $-1.95_{.63}$ | $-4.80_{.76}$ | $-3.26_{.80}$ | $-2.53_{.86}$ | $-1.53_{.95}$ |
| $0.50^{b1}$ | $-19.34_{.10}$ | $-14.88_{.13}$ | $-12.52_{.19}$ | $-10.08_{.27}$ | $-8.80_{.23}$ | $-6.85_{.27}$ | $-5.12_{.33}$ | $-4.25_{.41}$ | $-5.98_{.46}$ | $-4.00_{.50}$ | $-2.47_{.56}$ | $-1.92_{.64}$ | $-2.88_{.77}$ | $-2.42_{.81}$ | $-1.53_{.87}$ | $-0.97_{.96}$ |
| $0.75^{b1}$ | $-16.78_{.11}$ | $-14.22_{.15}$ | $-12.32_{.21}$ | $-10.90_{.49}$ | $-6.95_{.25}$ | $-6.56_{.29}$ | $-5.96_{.35}$ | $-4.11_{.43}$ | $-4.44_{.47}$ | $-2.67_{.51}$ | $-2.28_{.57}$ | $-1.89_{.66}$ | $-2.17_{.78}$ | $-2.26_{.82}$ | $-1.33_{.89}$ | $-0.37_{.97}$ |
| $1.00^{b1}$ | $-16.39_{.14}$ | $-13.61_{.17}$ | $-11.36_{.23}$ | $-9.27_{.31}$ | $-7.35_{.27}$ | $-5.38_{.31}$ | $-4.83_{.37}$ | $-3.56_{.45}$ | $-4.64_{.49}$ | $-2.38_{.53}$ | $-2.41_{.60}$ | $-1.73_{.68}$ | $-2.53_{.81}$ | $-2.76_{.85}$ | $-0.54_{.91}$ | $0.00_{1.0}$ |

**ResNet110**

| | $0.25^{b3}$ | | | | $0.50^{b3}$ | | | | $0.75^{b3}$ | | | | $1.00^{b3}$ | | | |
|---|---|---|---|---|---|---|---|---|---|---|---|---|---|---|---|---|
| | $0.25^{b2}$ | $0.50^{b2}$ | $0.75^{b2}$ | $1.00^{b2}$ | $0.25^{b2}$ | $0.50^{b2}$ | $0.75^{b2}$ | $1.00^{b2}$ | $0.25^{b2}$ | $0.50^{b2}$ | $0.75^{b2}$ | $1.00^{b2}$ | $0.25^{b2}$ | $0.50^{b2}$ | $0.75^{b2}$ | $1.00^{b2}$ |
| $0.25^{b1}$ | $-12.64_{.09}$ | $-8.63_{.12}$ | $-7.51_{.18}$ | $-7.00_{.26}$ | $-6.07_{.23}$ | $-6.85_{.26}$ | $-1.43_{.32}$ | $-0.69_{.40}$ | $-3.48_{.45}$ | $-1.51_{.49}$ | **$+0.21_{.55}$** | $-1.72_{.63}$ | $-2.28_{.77}$ | $-1.45_{.81}$ | **$+2.02_{.87}$** | **$+0.29_{.95}$** |
| $0.50^{b1}$ | $-10.26_{.10}$ | $-8.44_{.13}$ | $-6.12_{.19}$ | $-3.58_{.27}$ | $-4.57_{.24}$ | $-2.16_{.27}$ | $-0.54_{.33}$ | $-0.49_{.41}$ | $-1.45_{.46}$ | $-0.86_{.50}$ | $-0.33_{.56}$ | **$+2.58_{.64}$** | $-10.51_{.78}$ | $-0.64_{.82}$ | **$+1.64_{.88}$** | **$+0.37_{.96}$** |
| $0.75^{b1}$ | $-9.16_{.11}$ | $-8.88_{.15}$ | $-5.68_{.21}$ | $-4.51_{.29}$ | $-3.45_{.25}$ | $-0.96_{.29}$ | $-3.59_{.34}$ | **$+0.01_{.43}$** | $-1.80_{.48}$ | $-2.92_{.51}$ | $-0.24_{.57}$ | **$+0.43_{.66}$** | **$+0.26_{.80}$** | $-1.00_{.83}$ | **$+2.13_{.89}$** | **$+1.05_{.97}$** |
| $1.00^{b1}$ | $-9.23_{.13}$ | $-7.89_{.17}$ | $-5.35_{.23}$ | $-4.62_{.31}$ | $-3.66_{.27}$ | $-1.84_{.31}$ | $-1.20_{.37}$ | $-0.23_{.45}$ | $-1.79_{.50}$ | **$+2.26_{.54}$** | **$+2.50_{.59}$** | **$+3.58_{.68}$** | $-3.03_{.82}$ | $-1.47_{.85}$ | **$+1.29_{.91}$** | $0.00_{1.0}$ |

Table 7: **Accuracy & Memory on CIFAR100.** Each subtable lists all derived models for the denote table heading seed convnet.

