# OpenReview forum: "CNN Compression and Search Using Set Transformations with Width Modifiers on Network Architectures"
_ICLR.cc/2023/Conference — Submitted to ICLR 2023_

### Official Review · Reviewer_yX2W · 2022-10-24

**Confidence:** 4
**Clarity, Quality, Novelty And Reproducibility:** This paper has poor quality, clarity …
**Correctness:** 2
**Technical Novelty And Significance:** 1
**Empirical Novelty And Significance:** 1
**Recommendation:** 1

**Strength And Weaknesses:**

---
Strengths:
- This paper targets a critical problem: designing efficient neural architecture without much human effort and computational resources.

---
Weaknesses:
- The proposed method is essentially a grid search over a simplified design space. The main contribution of this paper lies in partitioning layers into segments. However, the authors have provided no insights into why layers with the same feature map size should have the same width multiplier (i.e., pruning ratio). The current solution appears like an arbitrary design space simplification without proper justifications.
- The paper is hard to follow. The algorithm boxes in Section 3 are composed of many sub-procedure (function) calls, which I find hard to understand without more clarification. Though the authors claim that they base their method on filter pruning, the proposed algorithm seems to retrain all the candidate networks from scratch without inheriting the weights from the original model.
- This paper does not provide any baselines in its experimental evaluation. As the proposed method is highly related to pruning and neural architecture search, it is more than necessary to include some numbers of related baselines.
- The experimental results are all on small-scale benchmarks, which is less representative. It would be essential to include some results on large-scale datasets, such as ImageNet. The authors claim that their search cost is much lower than other methods, which is actually because the dataset they are using is much smaller.

---

**Summary Of The Paper:**

This paper studies efficient neural network design for faster inference. The authors propose Structured Coarse Block Pruning (SCBP) that first partitions all layers into segments, then assigns a width multiplier to each stage, and finally explores the best configuration by grid search. The proposed SCBP delivers 20% faster and 50% smaller models on CIFAR-10 and CIFAR-100 benchmarks.

**Summary Of The Review:**

My recommendation is based on the limited novelty, insufficient evaluation, and poor writing of this paper. Therefore, I think it is not ready for publication in its current shape.

---

### Official Review · Reviewer_aX8y · 2022-10-30

**Confidence:** 4
**Correctness:** 1
**Technical Novelty And Significance:** 1
**Empirical Novelty And Significance:** 1
**Recommendation:** 3

**Clarity, Quality, Novelty And Reproducibility:**

The paper is clearly written and the method is reproducible. There is very little novelty in the proposed method as it amounts to manual CNN design based on a seed architecture.

**Strength And Weaknesses:**

The method proposed in the paper requires retraining every variation from scratch which is not network compression but simply architecture search. Since there is very little information proposed to guide this architecture search the paper performs either brute force architecture search (using a grid search) or manual architecture search which is simply CNN training.

To strengthen the paper the authors should compare with network compression techniques (such as the ones listed in the related work section) and show whether the proposed method finds architectures with specific efficiency characteristics more efficiently than network compression.

**Summary Of The Paper:**

The authors propose a systematic method of creating variations of a convolutional architecture by reducing the number of filters at certain groups of layers. Subsequently training and evaluating these new architectures shows that some are significantly more efficient that the initial "seed" architecture achieving similar accuracy scores with significantly less computational resources.

**Summary Of The Review:**

Based on the lack of comparison with baselines as well as the low novelty I propose rejection.

---

### Official Review · Reviewer_PfGt · 2022-10-31

**Confidence:** 4
**Clarity, Quality, Novelty And Reproducibility:** The work is not of good quality, nove…
**Correctness:** 2
**Technical Novelty And Significance:** 1
**Empirical Novelty And Significance:** 2
**Recommendation:** 3

**Strength And Weaknesses:**

The method proposed is effective and not hard to follow.
However, it brings limited new things/insights to me.
1. The authors claim the core of its method is the segmentation and the c-ratio set.
However, is the segmentation really needed?
The authors claim that the segmentation can reduce the search cost by binning similar layers together in a bucket/segment. It appears to me that this is very trivial.
2. The c-ration, in the paper, is also trivial to set.
3. Therefore, the framework is a bit of engineering with very limited research insight.
4. A previous work, AutoSlim[1], associates a ratio to each single layer, and applies algorithm to automatically learn the ratio, which should be discussed and compared.
5. The work does not compare to other methods.
6. The experiment evaluation is also limited on only CIFAR.
7. The authors claim there is no retrain cycle. However, after the compression, it requires to train and evaluated the candidates' architectures. This step is 'retrain' and would cost a lot of time.
8. The resulted network size and FLOPS are not reported.
9. The writing should be largely improved with many typos.

Reference:
[1] https://arxiv.org/abs/1903.11728

**Summary Of The Paper:**

This paper demonstrates SCBP, a framework to compress convolutional networks, by first segmenting the layers into superblocks and then associate them with different compression ratios. The SCBP framework constructs a pool of models with different attributes that meet different needs.

**Summary Of The Review:**

Overall, I think the work lacks novelty, research insights, experiments and analysis. It is not appropriate to be accepted in its current form.

---

### Decision · Program_Chairs · 2023-01-20

**Decision:**

Reject

**Justification For Why Not Higher Score:**

Compressing neural networks is a well-established area of research but there are no comparisons against previous work.

**Justification For Why Not Lower Score:**

NA

**Metareview: Summary, Strengths And Weaknesses:**

This paper performs a grid search over different pruning parameters to prune CNNs. While this paper targets an important problem, the AC shares the reviewers' concerns regarding novelty, clarity, and the fact that it is not compared against any baselines.